# Metastatic Breast Cancer Presenting as Acute Appendicitis: A Rare Case Study and Review of Current Knowledge

**DOI:** 10.3390/diagnostics13243657

**Published:** 2023-12-13

**Authors:** Nenad Markovic, Bojan Stojanovic, Ivan Jovanovic, Bojan Milosevic, Marko Spasic, Ivan Radosavljevic, Natasa Zdravkovic, Dragce Radovanovic, Bojana S. Stojanovic, Marija Spasojevic, Marina Jovanovic, Zeljko Todorovic, Mladen Pavlovic, Snezana Sretenovic, Milos Z. Milosavljevic, Milica Dimitrijevic Stojanovic

**Affiliations:** 1Department of Surgery, Faculty of Medical Sciences, University of Kragujevac, 34000 Kragujevac, Serbia; dr.nenadmarkovic@yahoo.com (N.M.); bojan.stojanovic01@gmail.com (B.S.); drbojanzm@gmail.com (B.M.); drmspasic@gmail.com (M.S.); drakce_5@hotmail.com (D.R.); mladenlucky@gmail.com (M.P.); 2Center for Molecular Medicine and Stem Cell Research, Faculty of Medical Sciences, University of Kragujevac, 34000 Kragujevac, Serbia; ivanjovanovic77@gmail.com; 3Department of Internal Medicine, Faculty of Medical Sciences, University of Kragujevac, 34000 Kragujevac, Serbia; natasasilvester@gmail.com (N.Z.); marinna034@gmail.com (M.J.); sretenovicsnezana@yahoo.com (S.S.); 4Department of Pathophysiology, Faculty of Medical Sciences, University of Kragujevac, 34000 Kragujevac, Serbia; bojana.stojanovic04@gmail.com; 5Department of Pathology, Faculty of Medical Sciences, University of Kragujevac, 34000 Kragujevac, Serbia; spasojevicmarija89@gmail.com (M.S.); m.milosavljevic77@gmail.com (M.Z.M.); milicadimitrijevic@yahoo.com (M.D.S.)

**Keywords:** metastatic breast cancer, acute appendicitis, invasive breast carcinoma of no special type, CDK4/6 inhibitors, hormone therapy, oncological diagnosis, immunohistochemical phenotype, clinical management

## Abstract

This manuscript discusses a rare case of acute appendicitis caused by metastasis from invasive breast carcinoma of no special type in a 70-year-old female previously diagnosed with breast cancer. It delves into the diagnostic challenges and management complexities of such unusual clinical presentations. The paper includes an analysis of 19 documented cases, enriching the understanding of metastatic patterns and treatment strategies in breast cancer. It underlines the importance of considering a history of malignancy when diagnosing acute abdominal conditions and emphasizes a comprehensive approach in interpreting diagnostic imaging in patients with past oncological issues to effectively manage metastatic breast cancer exhibiting atypical manifestations.

## 1. Introduction

Acute appendicitis is frequently encountered in emergency departments, primarily resulting from luminal obstruction at the appendicular base due to factors like fecaliths or lymphoid hyperplasia [1]. Malignant tumors account for less than 1% of these cases [2]. In exceptionally rare instances, acute appendicitis can arise due to metastases from extra-abdominal neoplasms, including breast cancer [3].

Breast cancer is the most prevalent cancer among women globally, comprising about 25% of all cancer cases, with an estimated 2.3 million new diagnoses annually [4]. It holds the highest incidence rate among women, with 127.5 cases per 100,000 per year, as reported by the Surveillance, Epidemiology, and End Results (SEER) program [5]. Invasive breast carcinoma of no special type (IBC NST), the most common breast cancer type, typically metastasizes to bones, lungs, liver, and other common sites [6]. However, its spread to the appendix is a rarity and is documented in only a few instances in the literature [3]. 

Our case report emphasizes the necessity of considering metastatic disease, especially in patients with a history of malignancy, as a potential cause of acute appendicitis. We aim to shed light on the diagnostic challenges, clinical manifestations, and management strategies of this rare condition. This report, enriched with a review of the literature and our analysis of 19 documented cases, seeks to enhance the understanding and treatment approaches for patients with metastatic breast carcinoma to the appendix, ultimately striving for improved patient outcomes.

## 2. Case Presentation

A 70-year-old female patient presented to the emergency department complaining of migratory abdominal pain. On clinical examination, the patient exhibited classical signs of acute appendicitis, including tenderness in the right iliac fossa, rebound tenderness, and guarding.

Delving into her medical history, it was revealed that in 2013, she was diagnosed with grade II IBC NST carcinoma of the breast. Post-diagnosis, she underwent surgical resection, and her post-operative course included treatment with the Fluorouracil, Adriamycin (Doxorubicin), and Cyclophosphamide (FAC) chemotherapy regimen, followed by tamoxifen maintenance. Subsequent checkups did not indicate any recurrence or persistence of the disease.

An abdominal ultrasound was performed, supplemented with routine blood tests to elucidate the cause of her symptoms and verify the clinical suspicion of acute appendicitis. The abdominal ultrasound revealed a dilated, non-compressible appendix located in the right iliac fossa, measuring approximately 10 mm in diameter. The appendiceal wall appeared edematous and hypoechoic, consistent with sonographic findings of acute appendicitis. Adjacent to the appendix, there was increased echogenicity of the surrounding adipose tissue, suggestive of inflammation. Additionally, reactive lymph nodes measuring approximately 7 mm in diameter were identified, along with the presence of loculated free fluid in the vicinity. 

Laboratory investigations revealed findings consistent with an acute inflammatory response. A complete blood count showed leukocytosis, with a white blood cell (WBC) count elevated at 10.14 × 10^9^/L. A detailed differential count highlighted a significant shift to the left, with granulocytes, predominantly neutrophils, comprising 73.20% of the total leukocyte count. Additionally, serum inflammatory markers were assessed, revealing a markedly elevated C-reactive protein (CRP) level at 80.90 mg/L. Other routine biochemical investigations were within normal limits, and no other significant abnormalities were identified in the laboratory profile.

Given the clinical and diagnostic findings, an urgent appendectomy was indicated. Upon commencing the lower midline laparotomy, the surgeons identified a markedly inflamed and hyperemic appendix. It was edematous, with visible engorged vessels over its serosal surface, but notably, there was no evidence of rupture (Figure 1). The periappendiceal region showed localized inflammatory changes, with the surrounding tissues appearing erythematous and slightly adhesive. No pus or abscess was observed. Additionally, the rest of the intra-abdominal structures, including the cecum, ileum, and adjacent tissues, were inspected and appeared normal without any other pathomorphological alterations. Following meticulous dissection, the inflamed appendix was excised. The operative site was then irrigated with warm saline to ensure thorough cleansing, and a drainage was placed to preemptively manage any potential postoperative complications. 

Upon histopathological examination of the excised appendix specimen, multiple discrete metastatic deposits were identified within the appendiceal wall (Figure 2). These deposits exhibited architectural and cytological features consistent with a moderately differentiated IBC NST carcinoma of breast origin. The neoplastic cells displayed a cohesive growth pattern, with cells forming duct-like structures in certain areas. The nuclei of these cells were moderately pleomorphic with prominent nucleoli.

To further characterize the origin and nature of these metastatic deposits, immunohistochemical studies were conducted (Figure 2). The tumor cells demonstrated strong positivity for cytokeratin 7 (CK7), E-cadherin, and GATA binding protein 3 (GATA 3), markers commonly associated with breast origin. Additionally, the cells exhibited strong estrogen receptor (ER) expression, quantified at 70% with an intensity score of +++ and a combined score of 7. Progesterone receptor (PR) was also positive, with 10% of the cells staining with an intensity of ++, resulting in a combined score of 4. Importantly, the tumor cells were negative for human epidermal growth factor receptor 2 (HER2) (scored at 0), suggesting the absence of HER2/neu gene amplification. Additionally, the markers caudal type homeobox 2 (CDX2) and CK20, which are typically associated with gastrointestinal and colorectal origins, were also negative, further corroborating the breast origin of the metastatic deposits. The TNM classification of the tumor was T2N0M0.

The post-operative course was uneventful, and the patient was discharged on the fifth post-operative day. Subsequently, her case was reviewed at a Multidisciplinary Team Meeting, where it was decided to initiate treatment with cyclin-dependent kinase 4/6 (CDK 4/6) inhibitors and letrozole, based on her cancer profile and surgical findings. Six months following her surgical procedure, the patient showed no signs of malignant disease recurrence.

## 3. Discussion

The unusual presentation of metastatic invasive breast carcinoma of no special type manifesting as acute appendicitis, as observed in our case, underscores the intricate and often unexpected pathways of breast cancer metastasis. Despite the commonality of breast cancer, metastasis to the appendix remains exceedingly rare, warranting a thorough re-evaluation of our understanding of metastatic patterns. This case highlights the critical importance of maintaining a high index of suspicion for metastatic involvement in patients with a history of malignancy presenting with acute abdominal symptoms. The finding of breast cancer metastasizing to the gastrointestinal tract, particularly to an organ as infrequently involved as the appendix, challenges the conventional clinical paradigms and underscores the necessity of considering a wide differential diagnosis. Our case, in conjunction with the previously documented instances, contributes to a growing body of evidence that suggests a more complex behavior of breast cancer metastases than previously understood.

### 3.1. Metastatic Involvement of the Appendix: A Rare Occurrence

While primary tumors of the appendix, such as those detailed in certain case studies [7], are a distinct category of appendiceal malignancies, isolated metastasis to the appendix from other cancers is infrequent and primarily arises from peritoneal seeding [8]. Histologically, these metastatic cancers typically exhibit a pattern of gradual serosal invasion, while often sparing the mucosal layer [9]. In patients with a history of cancer, the prevalence of benign appendicular conditions, such as acute appendicitis and primary appendiceal tumors, far exceeds that of metastatic appendiceal tumors. This rarity makes clinical differentiation challenging. Common primary origins of appendiceal metastases are the ovary, colorectum, and stomach, with the digestive tract being a less frequent site for isolated metastasis, often accompanied by disseminated disease in other organs [8]. Acute appendicitis, in these cases, may be the sole clinical manifestation directly attributable to the metastatic appendicular tumors [8]. The symptoms of early acute appendicitis can be ambiguous, further complicated by factors like prior radiation or chemotherapy and immune compromise in these patients, often leading to delayed diagnosis [10].

In a comprehensive study by Connor et al. [11], a retrospective review of 7970 appendectomies performed over 16 years revealed that only 0.9% (74 patients) had appendiceal tumors. Among these, 12 were benign, 42 were carcinoids, and 20 were malignant, with acute appendicitis being the most common presentation in 49% of these cases [11]. Notably, only 11 patients had secondary malignancies involving the appendix, with 55% of these cases originating from primary colorectal cancer. Management strategies varied widely, ranging from a simple appendectomy to a more extensive right hemicolectomy [11]. The primary origins of these metastatic appendiceal tumors included breast, lung, stomach, and colon, and the interval between primary cancer diagnosis and appendicitis onset ranged from immediate to as long as 6 years [12]. 

### 3.2. Breast Cancer Metastasis: Patterns and Rarity

Breast cancer patients face a varied risk of developing secondary lesions, with the incidence of metastasis depending on several factors, including the stage at diagnosis and the cancer type. While approximately 6% of women in the United States are diagnosed with metastatic breast cancer initially, a significant proportion, up to 30%, of those initially diagnosed with early-stage breast cancer may eventually progress to metastatic disease [13,14]. These metastases can manifest either synchronously, occurring simultaneously with the primary tumor in roughly 20% of cases, or metachronously, developing later in approximately 80% of cases, typically after a median follow-up period of 7 years [15]. The most prevalent sites for primary breast cancer metastasis include regional lymph nodes, bones, liver, lung, brain, and skin [16]. However, metastasis to the gastrointestinal (GI) tract, especially to the appendix, is exceedingly rare [5].

Invasive breast carcinoma of no special type, the most common form of breast cancer, generally metastasizes to more conventional sites, like bones, lungs, and liver [11]. The occurrence of breast carcinoma metastasizing to the appendix is very sparsely documented in the medical literature [17]. Despite its infrequency, this pattern of metastasis is an important consideration in breast cancer patients presenting with symptoms indicative of acute appendicitis, as it can profoundly impact both the diagnostic approach and the management strategy. This rare metastatic pathway underscores the need for a broad differential diagnosis in breast cancer survivors presenting with gastrointestinal complaints.

Moreover, brain metastases in breast cancer patients are significant, being the second most common cause of brain metastases in the United States after lung cancer. These metastases occur in 15% to 24% of women with metastatic breast cancer. The survival rate for breast cancer that metastasizes to the brain is notably lower, with survival times varying between 3 and 36 months. The 5-year survival rate for brain metastasis is only 1.51% [18].

### 3.3. Factors Influencing Metastatic Spread in Breast Carcinoma

The metastatic patterns of breast carcinoma are influenced by two key factors: estrogen receptor (ER) status and the cancer’s histopathology, whether ductal or lobular [19]. ER-positive cancers tend to have a higher propensity to metastasize to the bone, whereas ER-negative tumors are more likely to result in visceral metastases [20]. Lobular carcinoma, in particular, shows a predilection for metastasizing to the gastrointestinal (GI) and gynecologic systems as well as to the peritoneum and retroperitoneum [21]. This tendency can be partly attributed to the frequent occurrence of E-cadherin mutations in invasive lobular carcinoma, leading to a lack of expression of this crucial cell–cell adhesion molecule [22]. The compromised structural integrity of epithelial sheets due to the absence of E-cadherin is a critical factor in the pathogenesis of distant metastatic disease [23]. 

The case presented here is particularly exceptional given the unusual disease progression for an invasive breast carcinoma of no special type. The recurrence of this carcinoma in the appendicular wall, characterized by the absence of HER-2 staining and a profile of PR and ER positivity, positioned the patient at a lower risk for peritoneal and gastrointestinal recurrence of breast cancer. This unique case underscores the complexity of predicting metastatic pathways in breast cancer and highlights the need for personalized approaches to both diagnosis and treatment that consider the specific molecular and histopathological characteristics of each tumor.

### 3.4. Pathophysiology of Appendiceal Metastasis in Breast Cancer

The pathophysiological process leading to appendiceal metastasis involves the deposition of metastatic cells within the serosa of the appendix and subsequent infiltration towards the lumen, which can lead to luminal obstruction [24]. This obstruction is the primary cause of acute inflammation in the appendix. On a microscopic level, as depicted in Figure 2, metastatic tumor cells are observed abundantly in the submucosa, muscularis mucosa, and subserosa. These cell accumulations contribute to luminal constriction, predisposing the appendix to acute inflammation and, in severe cases, perforation.

Distal to the site of this tumor-induced obstruction, characteristic pathological changes occur in the appendiceal wall, such as edema and infiltration by polymorphonuclear leukocytes [25]. If this unchecked inflammation continues, it may escalate into more severe complications, like abscess formation, perforation, and potentially perforation peritonitis [25]. This progression underscores the importance of early detection and intervention in cases where appendiceal metastasis is suspected, particularly in patients with a history of breast cancer. The pathophysiology of appendiceal metastasis highlights the complex interplay between tumor biology and local tissue responses, emphasizing the need for a comprehensive approach to diagnosis and management.

### 3.5. Clinical Presentation and Diagnosis of Metastasis to the Appendix

Patients with metastasis to the appendix, akin to the presentation in our case, can exhibit symptoms characteristic of acute appendicitis. These include nausea, vomiting, anorexia, and, specifically, right lower abdominal pain [24]. In our reported case, the patient presented with migratory abdominal pain, accompanied by classical signs of acute appendicitis, such as tenderness in the right iliac fossa, rebound tenderness, and guarding, mirroring these common clinical manifestations.

During physical examination, these patients may show signs such as fever and localized right iliac fossa tenderness, which were also observed in our case [24]. It is noteworthy that leukocytosis may not always be present, adding a diagnostic challenge [24]. In terms of imaging, CT scans of the abdomen and pelvis serve as the primary radiological investigation. The salient features of acute appendicitis on CT scans, like a thickened appendiceal wall and periappendiceal fat stranding, align with the findings observed in our patient’s ultrasound, showing a dilated, non-compressible appendix and an edematous appendiceal wall [26]. Ultrasound and MRI are valuable alternatives when radiation exposure is a concern, as was the consideration in our diagnostic approach.

PET scanning is also a beneficial tool, particularly in patients with stage IV breast cancer who may not exhibit typical abdominal symptoms [27]. Differentiating between non-tumoral perforated appendicitis and perforated appendicular tumors can be complex due to overlapping clinical presentations [28].

### 3.6. Therapeutic Strategies for Breast Cancer Metastasis to the Appendix

In treating metastatic breast cancer (MBC) that has spread to the appendix, the chosen strategy is guided by the cancer’s immunohistochemical profile, particularly factors like estrogen receptor (ER) status and human epidermal growth factor 2 (HER2) status [29]. Hormone therapy is commonly prescribed for ER-positive cases, while anti-HER2 therapy is employed in HER2-positive cases [29]. Frequently, the approach for managing metastatic breast cancer is based on the immunohistochemical phenotype of the primary tumor removed during radical surgery. This reliance is often due to the challenges in securing biopsy samples from metastatic lesions [29]. If the initial therapeutic choice is ineffective, other treatments should be explored, or a pathological diagnosis of the metastatic lesion must be obtained through advanced techniques like computed tomography (CT)-guided biopsy, bone biopsy, or laparotomic biopsy, particularly since the immunohistochemical phenotype may differ between primary and metastatic lesions [29].

For cases like acute appendicitis secondary to breast tumor metastasis, the standard treatment is appendectomy. There is a lack of consensus on whether right hemicolectomy offers better oncological outcomes than appendectomy in stage IV breast cancer patients [9]. The selection of therapeutic strategies underlines the importance of tailored treatment plans, adapting to the unique challenges presented by metastatic breast cancer, especially when it manifests in less common locations, such as the appendix.

### 3.7. Role of Cyclin-Dependent Kinase 4/6 Inhibitors in Combination with Letrozole for Metastatic Breast Cancer

Cyclin-dependent kinase 4/6 (CDK4/6) inhibitors, when used in combination with hormone therapies such as letrozole, have emerged as a pivotal treatment strategy in hormone receptor-positive, human epidermal growth factor receptor 2 (HER2)-negative metastatic breast cancer [30]. The combination of ribociclib, a CDK4/6 inhibitor, with letrozole has demonstrated significant efficacy in improving overall survival rates for advanced breast cancer [31]. These inhibitors target the enzymes CDK4 and CDK6, which are crucial in cell division and are commonly overexpressed in breast cancer cells, providing a substantial therapeutic advantage [32].

The U.S. Food and Drug Administration (FDA)-approved CDK4/6 inhibitors—palbociclib, ribociclib, and abemaciclib—vary in their chemical and pharmacological characteristics. Palbociclib, the first of these agents to be approved, holds a significant market share in the U.S., with ribociclib and abemaciclib also being widely utilized [33]. The treatment approach used in our case, involving the combination of CDK 4/6 inhibitors with letrozole, aligns with the current therapeutic practices for managing hormone receptor-positive, HER2-negative metastatic breast cancer. This approach was instrumental in achieving a favorable outcome in our patient, highlighting the significance of selecting a treatment regimen based on the cancer’s specific immunohistochemical profile.

### 3.8. Chemotherapy-Induced Immunocompromise: Heightening Complications in Appendicitis

When appendicitis occurs in breast carcinoma patients with malignant metastases, the immunocompromised state induced by both the advanced cancer and chemotherapy plays a critical role. This immunocompromise leads to a late presentation of appendicitis symptoms and elevates the risk of severe complications, such as perforation [5]. The situation is further complicated by the fact that chemotherapy’s side effects, which often include nausea, vomiting, and abdominal pain, can closely mimic the signs and symptoms of acute appendicitis. Such symptomatic overlap frequently results in delayed diagnosis, adding to the complexity of managing these patients [10].

Highlighting the rare presentation of stage IV breast carcinoma in such clinical scenarios is crucial, as it underlines the need for physicians to be vigilant and suspect appendicitis in appropriate settings [5]. The risk of increased morbidity and mortality in these cases cannot be overstated. Additionally, treatment-related factors, like the use of corticosteroids, whether intermittent or continuous, can predispose patients to specific complications, such as neutropenic enterocolitis, further challenging the accurate and timely diagnosis of abdominal pain in cancer patients [34]. This underlines the importance of a nuanced approach to the diagnosis and management of abdominal conditions in cancer patients, particularly those undergoing chemotherapy.

### 3.9. Considerations for Prophylactic Appendectomy in Breast Cancer Patients

In light of the increased incidence of perforation among breast cancer patients with metastases to the appendix, the idea of a prophylactic appendectomy warrants consideration. This preventive surgical approach could be particularly beneficial for patients already undergoing invasive abdominal interventions, such as oophorectomy or other types of abdominal surgery [35]. However, it is important to note that, as of now, there are no definitive guidelines or universally accepted protocols advocating for routine prophylactic appendectomy in these circumstances.

The decision to perform a prophylactic appendectomy must be carefully weighed considering the patient’s overall health status, the risks associated with additional surgery, and the specific circumstances of their cancer diagnosis and treatment plan. This approach highlights the need for a personalized and strategic surgical plan tailored to the individual needs and risks of each patient.

### 3.10. Prognostic Implications of Appendiceal Metastasis in Breast Cancer

The prognosis for patients experiencing metastasis of breast cancer to the appendix, particularly those presenting with acute appendicitis, tends to be unfavorable. Studies indicate that the median survival following the diagnosis of secondary appendiceal tumors is approximately 22.6 months [8]. A contributing factor to this poor prognosis is the rarity of isolated metastasis to the appendix, coupled with the frequent presence of peritoneal metastases in such cases [8]. 

### 3.11. Appendicitis as a Presentation of Metastatic Breast Cancer: A Chronological Review of 19 Cases

In reviewing the 19 documented cases of metastatic breast cancer (MBC) causing appendicitis, including our recent presentation, a comprehensive understanding emerges of this rare but significant clinical phenomenon (Table 1). The cases span from 1946 to 2022, highlighting an enduring clinical relevance. Patient ages in these reports ranged from 35 to 73 years, with an average age of around 55 years, suggesting a higher incidence in the middle-aged to elderly population.

Histopathologically, the majority of these cases involved invasive ductal carcinoma, a common form of breast cancer. Notably, four cases specifically involved metastatic lobular carcinoma (Oldfield et al. [36], Pigolkin et al. [41], Dirksen et al. [42], and Numan et al. [5]), indicating a potential propensity for lobular carcinoma to metastasize to the appendix. In terms of hormone receptor status, when reported, ER positivity was predominant and found in cases like Hughes et al. [9], Kwan et al. [29], and Vincent De Pauw et al. [3]. PR positivity and HER2 negativity were also commonly observed, underscoring the hormonal influence in these metastatic events.

The clinical presentations predominantly featured acute appendicitis, with several cases reporting perforation, a complication adding to the urgency and severity of the condition. Surgical interventions varied, with most cases undergoing appendectomy, either laparoscopic or open, as seen in Vincent De Pauw et al. [3], Mori et al. [43], and Meenakshi Yeola et al. [24]. In contrast, more extensive procedures, like right hemicolectomy, were required in instances of extensive disease or perforation, as illustrated by Ng et al. [44].

The first documented case, reported by Oldfield in 1946 (Oldfield et al. [36]), set the precedent for understanding this manifestation of MBC. Since then, cases like those reported by Vincent De Pauw et al. [3] and Hughes et al. [9] have contributed to a growing recognition of the varied and often unexpected pathways of breast cancer metastasis.

Interestingly, the outcomes following surgical intervention varied significantly, ranging from uneventful recoveries to fatalities shortly after surgery, as in the case of Vincent De Pauw et al. [3]. 

For a comprehensive overview of the crucial aspects discussed in this manuscript, please refer to Table 2. This table concisely encapsulates the key elements, including the incidence, diagnostic challenges, therapeutic approaches, and prognostic implications associated with metastasis of breast carcinoma to the appendix. 

## 4. Conclusions

Acute appendicitis resulting from metastasis of invasive ductal breast carcinoma is a rare phenomenon. Despite the low incidence of peritoneal carcinomatosis in breast cancer, it is crucial for clinicians to be vigilant about this possibility, particularly in patients with a history of breast cancer, even after extended periods without recurrence. Such metastasis can present with various abdominal symptoms, including acute appendicitis. Hence, abdominal imaging is mandatory and needs to be meticulously interpreted within the context of the patient’s oncological history. Recognizing and appropriately managing such rare cases are key to improving patient outcomes in the complex landscape of metastatic breast cancer.

## Figures and Tables

**Figure 1 diagnostics-13-03657-f001:**
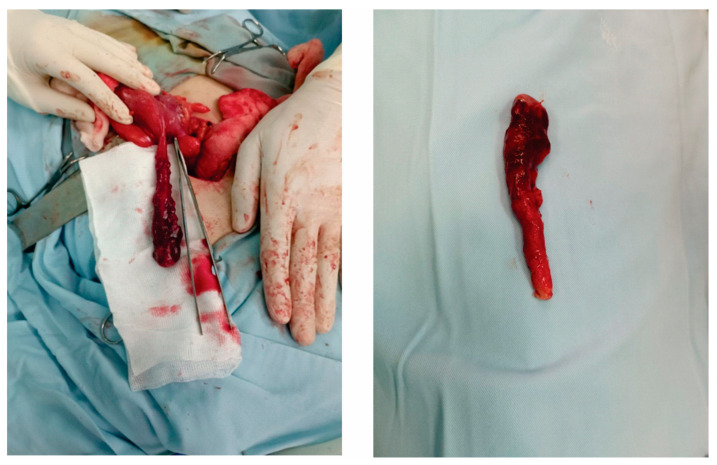
Intraoperative findings of the appendix. This figure depicts the intraoperative appearance of an appendix, characterized by marked inflammation and hyperemia. The appendix, measuring over 10 cm in length, shows significant edema with visibly engorged vessels on its serosal surface. Notably, despite the extensive inflammation, there is no evidence of rupture.

**Figure 2 diagnostics-13-03657-f002:**
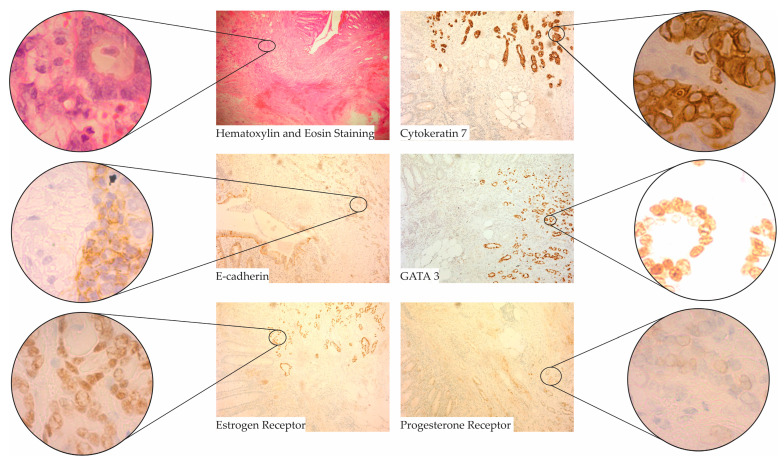
Histopathological characterization of metastatic breast carcinoma in the appendix. This figure illustrates the histological and immunohistochemical features of the metastatic breast carcinoma in the excised appendix specimen. The areas within the rectangles are magnified at ×40, while the areas within the circles are magnified at ×100. The hematoxylin and eosin (H&E)-stained section reveals the architectural and cytological patterns typical of moderately differentiated invasive breast carcinoma of no special type (IBC NST), with cohesive growth and duct-like structures. The immunohistochemical panel highlights the tumor cells’ strong positivity for cytokeratin 7 (CK7), E-cadherin, GATA binding protein 3 (GATA 3), estrogen receptor (ER), and progesterone receptor (PR), while corroborating the breast origin of the metastatic cells.

**Table 1 diagnostics-13-03657-t001:** Metastatic Breast Cancer Leading to Appendicitis: A Review of Documented Cases.

Author(s)	Year	Age/Sex	Interval BC-AA ^1^	Diagnosis	Surgery Type	Cancer Type	Hormonal Status	Treatment	Outcome
Oldfield M C [36]	1946	40 y F	3 years	Appendicitis	Open appendectomy	IBC NST	Not specified	N/A	N/A
Latchis et al. [37]	1966	45 y F	6 years	Appendicitis	Open appendectomy	Invasive lobular carcinoma	Not specified	N/A	N/A
Burney et al. [38]	1974	35 y F	1 year 4 months	Perforated appendicitis	Open appendectomy	N/A	N/A	N/A	Survived 4 months after appendectomy
Burney et al. [38]	1974	73 y F	3 years	Perforated appendicitis	Open appendectomy	N/A	N/A	N/A	Survived 1 month after appendectomy
Solis et al. [39]	1986	60 y F	5 years	Appendicitis	Open appendectomy	IBC NST	Not specified	N/A	N/A
Maddox P R [35]	1990	65 y F	5 years	Appendicitis	Right hemicolectomy	IBC NST	Not specified	N/A	N/A
Varga et al. [40]	2005	45 y F	Simultaneous	Perforated appendicitis	Appendectomy	IBC NST	Not specified	N/A	N/A
Pigolkin et al. [41]	2008	60 y F	18 years	Appendicitis	Appendectomy	Cancer type not specified	ER positive, PR positive	N/A	N/A
Dirksen et al. [42]	2010	76 y F	Simultaneous	Perforated appendicitis	Open appendectomy	Invasive lobular carcinoma	ER positive, PR negative, HER-2 negative	N/A	N/A
Mori et al. [43]	2016	56 y F	2 years 5 months	Appendicitis	Laparoscopic appendectomy	Invasive lobular carcinoma	ER positive, PR positive, HER-2 positive	Trastuzumab-DM1	Well controlled
Kwan et al. [29]	2016	70 y F	9 months	Appendicitis	Laparoscopic appendectomy	IBC NST	ER positive, PR negative, HER2 negative	Hormonal therapy (anastrozole)	Alive 9 months post-operation with no recurrence
Araujo et al. [12]	2018	37 y F	Simultaneous	Perforated appendicitis	Open appendectomy, hysterectomy, oophorectomy	IBC NST	ER negative, PR negative, HER-2 positive	Cyclophosphamide, fluorouracil, adriamycin therapy	N/A
Ng et al. [44]	2018	59 y F	2 years	Perforated appendicitis	Open right hemicolectomy	IBC NST	ER negative, PR positive, HER-2 positive	Chemotherapy and radiotherapy	N/A
Numan et al. [5]	2019	44 y F	3 years	Small bowel obstruction and appendicitis	Open ileocecectomy, appendectomy, adhesiolysis	Invasive lobular carcinoma	ER positive, PR positive, HER-2 negative	N/A	N/A
Vincent De Pauw et al. [3]	2020	64 y F	20 years	Appendicitis	Laparoscopic appendectomy	Invasive ductal carcinoma	Progesterone receptor positive, ER weaker positive, HER-2 negative	Refused anticancer treatment	Died a day after surgery
Meenakshi Yeola et al. [24]	2021	59 y F	Simultaneous	Perforated appendicitis	Laparoscopic appendectomy	Infiltrating ductal carcinoma with triple-negative status	Adriamycin, Cyclophosphamide, Paclitaxel	Alive 2 years post-operation with no recurrence	
David T Khalil et al. [17]	2022	55 y F	6 years	Suppurative inflammation, appendix perforated	Open cecectomy	Invasive ductal carcinoma	ER positive, PR negative, HER-2 negative	Hormonal therapy (exemestane)	N/A
Hughes et al. [9]	2022	51 y F	12 years	Perforated appendicitis	Laparoscopic appendectomy	Invasive lobular carcinoma	ER positive, PR positive, HER-2 positive	Hormone therapy and targeted therapy	N/A
Present Case	2023	70 y F	9 years	Acute appendicitis	Open appendectomy	Moderately differentiated ductal carcinoma of the breast	Strong ER expression (70%), PR positive (10%), HER2 negative	Continued treatment with CDK 4/6 inhibitors and letrozole	Free from malignant disease recurrence 6 months post-surgery

^1^ Note: “Interval BC-AA” denotes the time interval between breast cancer (BC) diagnosis and acute appendicitis (AA) occurrence. “N/A” indicates that the information was not available or not applicable.

**Table 2 diagnostics-13-03657-t002:** Key Findings and Considerations in Metastasis of Breast Carcinoma to the Appendix.

Aspect	Summary of Key Findings and Considerations
Incidence and Rarity	Metastasis of breast cancer to the appendix, particularly from invasive ductal carcinoma, is extremely rare.
Diagnostic Challenges	Diagnosis is challenging due to overlapping symptoms with typical acute appendicitis and the rarity of the condition.
Immunohistochemical Profile	Treatment decisions are often based on the immunohistochemical profile of the primary tumor, especially ER and HER2 status.
Metastatic Patterns	Breast carcinoma metastasizes to the appendix in a minority of cases, often involving the serosa and leading to luminal obstruction.
Therapeutic Approaches	Initial treatment typically involves surgery, primarily appendectomy. This may be followed by adjuvant therapy, and, in some metastatic cases, treatment includes CDK4/6 inhibitors combined with hormone therapy like letrozole.
Prognostic Implications	Prognosis is generally poor, with a median survival of 22.6 months after diagnosis of secondary appendiceal tumors.
Clinical Implications	Clinicians must maintain a high index of suspicion for metastatic disease in patients with a history of breast cancer presenting with abdominal symptoms.

## Data Availability

This article, being a case report and review, does not contain any primary data for sharing. The data discussed are derived from previously published studies and the patient’s medical records.

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
