# Peer review of "Metastatic Breast Cancer Presenting as Acute Appendicitis: A Rare Case Study and Review of Current Knowledge"

_diagnostics, 2023, doi:10.3390/diagnostics13243657_

Round 1

Reviewer 1 Report

Comments and Suggestions for Authors

This is an interesting and indeed rare case, but you do not need to call it rare or "rarity" throughout the whole manuscript. State it in the beginning with a reference. I the attached manuscript I have put in yellow a number of words and passages that you should omit. They are not contributing to the scientific value.

"Ductal" carcinoma is not the recommended term anymore. Refer to WHO Breast 5th edition. The entity is called carcinoma of no specific type (NST).

Give us the complete TNM classification of the tumour (line 61-65).

The images need to be replaced. The magnification is not large enough. Go to x 100. It is not possible to see the morphology on the submitted images.

Omit subsection 3.3

Describe the general symptoms of metastases to the appendix, not only with respect to BC (subsection 3.5)

Omit subsections 3.6 and 3.7. They are not contributory to a case presentation

Shorten subsection 3.9

Author Response

Dear Reviewer 1,

Thank you very much for your insightful and constructive feedback on our manuscript. We truly appreciate your guidance and have made the following revisions based on your suggestions:

Point: Overuse of the term "rarity" throughout the manuscript.

Response: We have reviewed and removed or modified most of the marked portions that overused the term "rarity," ensuring it's stated clearly at the beginning with appropriate reference.

Point: Using "Ductal" carcinoma term.

Response: As per your suggestion, we have replaced this term with "carcinoma of no specific type (NST)" throughout the text, in line with WHO Breast 5th edition recommendations.

Point: Complete TNM classification of the tumor.

Response: The TNM classification of the metastatic tumor in the appendix has been added to the manuscript.

Point: Image magnification.

Response: Alongside the existing images with 40x magnification, we have included new images at 100x magnification, providing a clearer view of the morphology and enhancing the visual representation of our findings.

Point: General symptoms of metastases to the appendix.

Response: Subsection 3.5 has been modified to initially describe the symptoms of all metastatic tumors in the appendix, not limited to breast cancer.

Point: Omitting subsections 3.3, 3.6, 3.7, and shortening subsection 3.9.

Response: We respectfully disagree with the suggestion to omit these sections. Subsection 3.3 provides crucial context on factors influencing metastatic spread in breast carcinoma, essential for understanding the case. Subsections 3.6 and 3.7 discuss therapeutic strategies and the role of specific inhibitors, which are central to our case's treatment approach. Additionally, subsection 3.9 on prophylactic appendectomy in breast cancer patients has been well-received by another reviewer, reinforcing its relevance and importance in the manuscript.

We believe these sections contribute significantly to the scientific value and understanding of the case. Their inclusion offers a comprehensive view of the various aspects surrounding metastatic breast cancer to the appendix.

Once again, thank you for your valuable contributions to our work.

Sincerely,

Ivan Radosavljevic, MD, PhD

Zeljko Todorovic, MD, PhD

Reviewer 2 Report

Comments and Suggestions for Authors

The review presents an intriguing case report of a rare breast cancer metastasis.

The introduction is great in its briefness and clarity.

Case presentation is written in a logical and comprehensive way.

I would suggest moving the histology data  (lines 111-128)  above the treatment decision (lines 106-110) and follow-up.

When describing immunohistochemistry, it would be interesting to compare the present receptor profile (ER, PR, HER2) with the one of a primary tumor.

The data of the Ki67 proliferation index would also be appreciated.

The discussion is structured well too, with correct references.

In my opinion, Section 3.2 would benefit from mentioning brain metastases, as the common cause of these patients' death.

Considerations for Prophylactic Appendectomy is a very interesting point.

Table 2 is a beautiful sum-up of a known data.

The Conclusions are clear and concise.

The reference list is up-to date.

Author Response

Dear Reviewer 2,

Thank you very much for your detailed and constructive feedback on our manuscript. We are grateful for your insights, which have greatly improved the clarity and depth of our report.

Point: Moving the histology data (lines 111-128) above the treatment decision (lines 106-110) and follow-up.

Response: We have rearranged the manuscript accordingly, placing the histology data before the treatment decision and follow-up. This change enhances the logical flow of the paper and provides better context for the treatment decisions.

Point: Comparing the present receptor profile (ER, PR, HER2) with that of the primary tumor.

Response: Unfortunately, we did not have access to the immunohistochemistry data of the primary tumor, preventing us from making this comparison.

Point: Mentioning brain metastases in Section 3.2.

Response: We have included information about brain metastases in the final paragraph of Section 3.2, emphasizing their significance as a common cause of mortality in these patients.

Thank you once again for your valuable contributions to our work.

Sincerely,

Ivan Radosavljevic, MD, PhD

Zeljko Todorovic, MD, PhD